# Data collected by a drone backpack for air quality and atmospheric state measurements during Pallas Cloud Experiment 2022 (PaCE2022)

David Brus[1], Viet Le[1], Joel Kuula[1] and Konstantinos Doulgeris[1]

[1]Finnish Meteorological Institute, Erik Palménin aukio 1, FI-00506, Helsinki, Finland

*Correspondence to*: David Brus (david.brus@fmi.fi)

**Abstract.**

A lightweight custom built drone backpack for air quality and atmospheric state variables measurements on top of consumer-grade drone was used during Pallas Cloud Experiment (PaCE) campaign's intensive operation period (IOP) between September 12th and October 10th, 2022. The drone backpack measurements include 63 vertical profile flights from two close by locations at Pallasjarvi lake and 12 inter-comparison flights against the reference instrumentation at Sammaltunturi station. The observations include aerosol number concentrations and size distributions, and meteorological parameters (temperature, relative humidity, pressure, wind speed and direction) up to 500 m above the ground level. The dataset has been uploaded to the common Zenodo PaCE 2022 community archive ( https://zenodo.org/communities/pace2022/ , last access: 5 June, 2025). The datasets in two formats NetCDF and CSV are available here: https://doi.org/10.5281/zenodo.14780929, Brus et al. (2025a) and https://doi.org/10.5281/zenodo.14778421, Brus et al. (2025b), respectively.

## 1 Introduction

Aerosol-cloud interactions and cloud microphysics are a weak point in all atmospheric models regardless of resolution (e.g. Morrison et al., 2020). It has been particularly challenging to accurately represent the Arctic climate system and Arctic clouds in regional and large-scale climate models (e.g. Sedlar et al. 2020). Cloud properties are sensitive to aerosols, which act as cloud condensation nuclei (CCN) and ice-nucleating particles (INP), especially in areas where the aerosol and CCN concentrations are low (Stevens et al. 2018). This makes the cloud properties in the Arctic and subarctic susceptible to anthropogenic pollution (e.g. Coopman et al. 2018; Liu & Li, 2019, Doulgeris et al. 2023). While there have been several large-scale efforts to characterize aerosol-cloud-climate interactions in the central and high Arctic (e.g. Abbatt et al. 2019; Pasquier et al. 2022), the subarctic zone has received relatively little attention. However, our previous research has shown that even the relatively clean air in Finnish subarctic is a complex mixture of potential aerosol precursors from various marine, biological and anthropogenic sources and that changes in anthropogenic emissions can have an impact on subarctic aerosol. For example, one of the largest sources of anthropogenic atmospheric $SO_2$ in the Finnish Lapland was the smelter industry in Kola Peninsula (Asmi et al. 2011, Kyrö et al. 2014, Jokinen et al. 2022). The relative contributions of different aerosol sources in the Arctic and subarctic can be expected to change in response to climate change (increasing temperatures, reduced snow

and ice cover, increasing biogenic emissions) and changes in human activity (changes in long-range transport and local activities e.g. shipping, mining, etc.). The implications of these rapid changes currently happening in the Arctic and subarctic on aerosol–radiation interactions and low-level clouds have remained elusive (Schmale et al. 2022).

The use of unmanned aircraft systems (UAS) to probe the atmosphere is stably increasing. The current UAS are relatively easy to deploy and have an advantage of frequent, high-resolution and affordable profiling of the atmosphere. The latest effort in

miniaturization of instrumentation and cost reduction of hardware, and availability of the consumer grade platforms allowed atmospheric research to increasingly expand to vertical dimension. Several atmospheric scientific UAS campaigns have focused on data collection of meteorological parameters, aerosols and clouds in the atmospheric boundary layer and higher latitudes (e.g. Altstädter et al., 2015; deBoer et al., 2020; Kral et al., 2021, Girdwood et al., 2022).

Several studies conducted vertical measurements of particulate matter concentrations and aerosol size distributions using

different variations of heavy lifting multicopters, equipped with off-shelf or custom build instrumentation, in environments with both low and very high loads of atmospheric pollutants (e.g. Zhu et al., 2019; Liu et al., 2020; Samad et al., 2020; Brus et al., 2021, Thivet et al., 2024). Over the last two decades, the Finnish Meteorological Institute (FMI) conducted extensive research on aerosol-cloud-interactions with focused Pallas Cloud Experiments (PaCE), focused on different phenomena, covering sources of precursors for new particle formation, aerosol chemistry, CCN activation, in-situ cloud measurements,

remote sensing and satellite observations. During PaCE 2022, several airborne platforms were deployed simultaneously to determine the ice nucleation particles (INP) concentrations, in-situ aerosol and cloud droplets number concentrations and turbulence measurements in atmospheric vertical profiles. Also concurrent measurements from the hilltop station are available as a reference for intercomparison, see Brus et al., (2025c) for details on the campaign setup.

In this study, we provide datasets of meteorological variables, aerosol concentrations, and size distributions collected during

intensive operation period (IOP) of Pallas Cloud Experiment (PaCE 2022) campaign between September 12[th] and October 10[th]. Our dataset offers a unique opportunity for the broader scientific community to better understand the vertical structure of near-surface aerosol particles in a subarctic environment, revealing their crucial role in influencing low-level stratiform cloud microphysical and radiative properties. In Section 2, we describe the drone measurement platform, the assembly of the backpack module with all sensors and their operational characteristics. Section 3 details the measurement sites, flight strategy

and presents the completed vertical profile measurements. Section 4 explains the dataset structure, quality control and assurance of data. Section 5 provides direct links to Zenodo dataset repository with netCDF and CSV files.

## 2 Description of platform, module and sensors

The FMI team operated a small consumer grade multicopter drone DJI Mavic 2 Pro with an on-top-of-drone mounted custom built measurement backpack. The drone backpack is an entry level air quality and atmospheric state measurement data

acquisition system built around the Raspberry Pi zero W microcomputer. It is intended to be used for air quality monitoring, vertical or horizontal profiling of atmosphere, or any educational application in the field of aerosols or meteorology. The backpack utilizes a custom-designed drone mounting and sensor attachment system. The housing of the backpack is 3d printed

from a white polyethylene terephthalate glycol (PETG) filament material to reflect the sun. The backpack is powered from a drone battery, and it is placed on top of the drone to minimize the propeller airflow on the particulate matter (PM) measurements to a high extent (e.g. Ghirardelli et al., 2023) while still provides enough air flow around the sensors measuring atmospheric state variables; temperature (T), relative humidity (RH) and pressure (P). The sensors are not force aspirated; the aspiration rather depends on drone vertical move and horizontal airflow for the air exchange around the sensors. Typically, lightweight and small size sensors were used: BME280 (T, RH, P, Bosch Sensortec) and SHT85 (T, RH, Sensirion AG).. The backpack also included a GNSS module (Dual BN-220, Beitian Co. Ltd.) to retrieve redundancy positioning, vertical and horizontal speed of the drone. The particulate matter (PM) sensor (OPC-N3, Alphasense Ltd., e.g. Sousan et al., 2016; Crilley et al., 2018; Hagan and Kroll, 2020) was mounted on top of the backpack cover by using zip ties, see Fig 1. The PM sensor did not include any extra inlet, nor drying system for aerosol flow. The inlet was horizontal heading towards the drone front. The sensors' measurement range and accuracy are provided in Table 1. The estimates of wind speed and direction were calculated from the drone attitude, speed and spatial orientation at each point of the flight. However, a third-party application was used for wind estimates, available at the Airdata.com portal with HD360 subscription. A proprietary wind algorithm version 2.2 and Mavic 2 pro aerodynamic profile were used for wind estimates within this manuscript. The Airdata.com application provided a wind map with maximum of 5 sec resolution which corresponded to approximately 10 – 15 m in our vertical profile measurements.  All sensor's variables were recorded in 1Hz resolution to the Rpi using dedicated simple Python scripts.

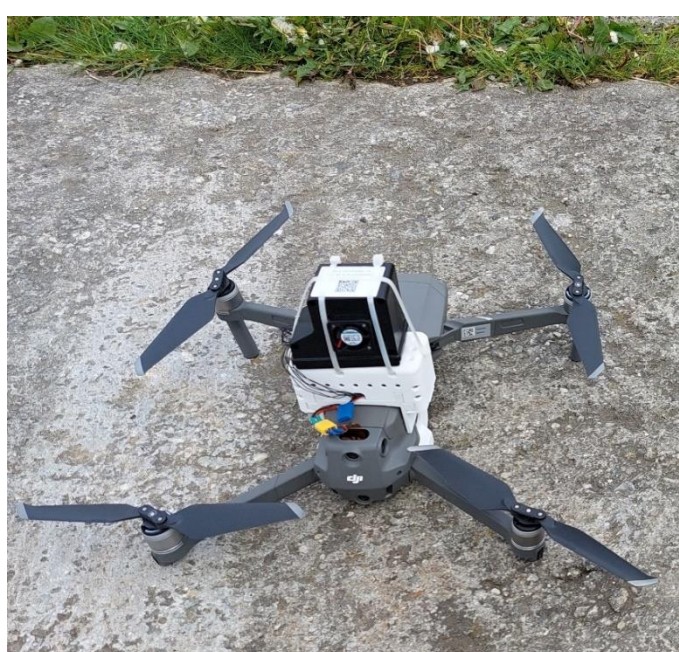

Figure 1. The DJI Mavic 2 pro equipped with a drone air quality backpack.

**Table 1** An overview of sensors and their operational characteristics provided by manufacturers. The details on sensors accuracy and tests could be found in following manufacturers datasheets: OPC-N3 (Alphasense Ltd., https://ametekcdn.azureedge.net/mediafiles/project/oneweb/oneweb/alphasense/products/datasheets/alphasense_opc-n3_datasheet_en_1.pdf?revision:29541b07-612a-42ba-b362-f41a48cf2e48, BME280 (Bosch Sensortec GmbH, https://www.bosch-sensortec.com/media/boschsensortec/downloads/datasheets/bst-bme280-ds002.pdf), SHT 85 (Sensirion AG, https://sensirion.com/resource/datasheet/sht85.

| Sensor | Resolution | Accuracy | Range | Response time |
|---|---|---|---|---|
| **OPC, model N3** | | | | |
| Particle conc. (cm$^{-3}$) | PM1, PM2.5, and PM10 mass concentrations 0.1 µg.m$^{-3}$ | | 0-10$^4$ part./s <br> 0.35-40 µm at 24 bins <br> Sample flow rate: 220 ml.min$^{-1}$ <br> Total flow rate: 5.5 l.min$^{-1}$ | 1 s |
| T(°C) | | | -10 to 50 °C | |
| RH (%) | | | 0-95% (non-condensing) | |
| **BME280** | | | | |
| T(°C) | 0.01 | ±0.5 °C | -40-85 °C | 1 s |
| RH (%) | <0.01 | ±3 % | 0-100 % | 1 s  at 25°C |
| Pressure (hPa) | 0.18 Pa | ±1 hPa | 300-1100 hPa | 6 ms |
| **SHT85** | | | | |
| T(°C) | 0.01 | ±0.1 °C | -40 -125°C | 5 s |
| RH (%) | 0.01 | ±1.5% | 0-100 % | 8 s at 25°C and 1m.s$^{-1}$ airflow |
| **GPS** <br> **Beitian Dual BN-220** | Horizontal 2 meters, vertical approx. 3 times horizontal | 0.1 ms$^{-1}$ | chipset 8030-KT <br> Frequency: GPS L1, GLONASS L1, BeiDou B1, SBAS L1, Galileo <br> Timing: 1µs synchronized to GPS time <br> Channels: 72 Searching Channel | Cold    start: 26s <br> Warm    start: 25s <br> Hot start: 1s |

## 3 Description of measurement sites and completed vertical profiles

The Pallastunturi Atmosphere-Ecosystem Supersite (https://en.ilmatieteenlaitos.fi/pallas-atmosphere-ecosystem-supersite) is part of European Research Infrastructures and networks including ACTRIS (https://www.actris.eu/), ICOS (https://www.icos-cp.eu/), EMEP and Global WMO GAW (https://wmo.int/activities/global-atmosphere-watch-programme-gaw), for details on research programs see e.g. Lohila et al., (2015). There are no main emission sources close by Pallas supersite. The drone flights during PaCE 2022 campaign were carried out within the FMI Arctic UAV base, the reserved airspace – temporary dangerous area (TEMPO D – Pallas). The airspace is authorized for beyond visual line-of-sight (BVLOS) operations covering a square region with a side length of approx. 14 km and an altitude ceiling limit FL80 (~2000 m AGL). The area is centered around the Sammaltunturi station, located above the tree line on the top of hill Sammaltunturi (67°58'24.0"N 24°06'56.3"E, 560 m ASL), approximately 300 m above the surrounding area. The station is commonly inside low-level clouds, especially during the fall season. The drone flights were conducted at two relatively nearby locations: the Pallasjärvi lake beach (68°01'23.2"N 24°09'48.8"E, 276 m MSL) and the UAV take-off/landing site next to main road 957 Pallaksentie (68°1'10.30"N 24°8'57.84"E, 304 m ASL). For details, see Figure 2 and the map in Brus et al., (2025c). The intercomparison flights against reference instrumentation were done at the Sammaltunturi station and next to the measurement tower of SMEAR 3 station at Kumpula campus in Helsinki (60°12.173'N 24°57.663'E).

During the IOP of PaCE 2022, we flown missions focusing on the profiling of meteorological parameters and aerosol particle in size range of 0.35 – 40 µm. In total, we completed 63 vertical profile flights: 42 flights the Pallasjärvi lake beach, 21 flights next to main road 957 Pallaksentie and 12 flights against the reference at Sammaltunturi station (see Figure 3). Only vertical profile data are included in the provided dataset. Our flight strategy was to conduct only high-resolution strictly vertical flights and reach the maximum altitude of the DJI Mavic 2 pro drone, 500 m AGL. The flights were conducted with OPC-N3 inlet always heading into wind. The flight missions were conducted by using DJI GO 4 software, and both ascent and descend rates were set the same to 1 m s$^{-1}$. Vertical profiling was aimed to be performed whenever the weather allowed it, except days with morning fog appearing, or precipitation at site.

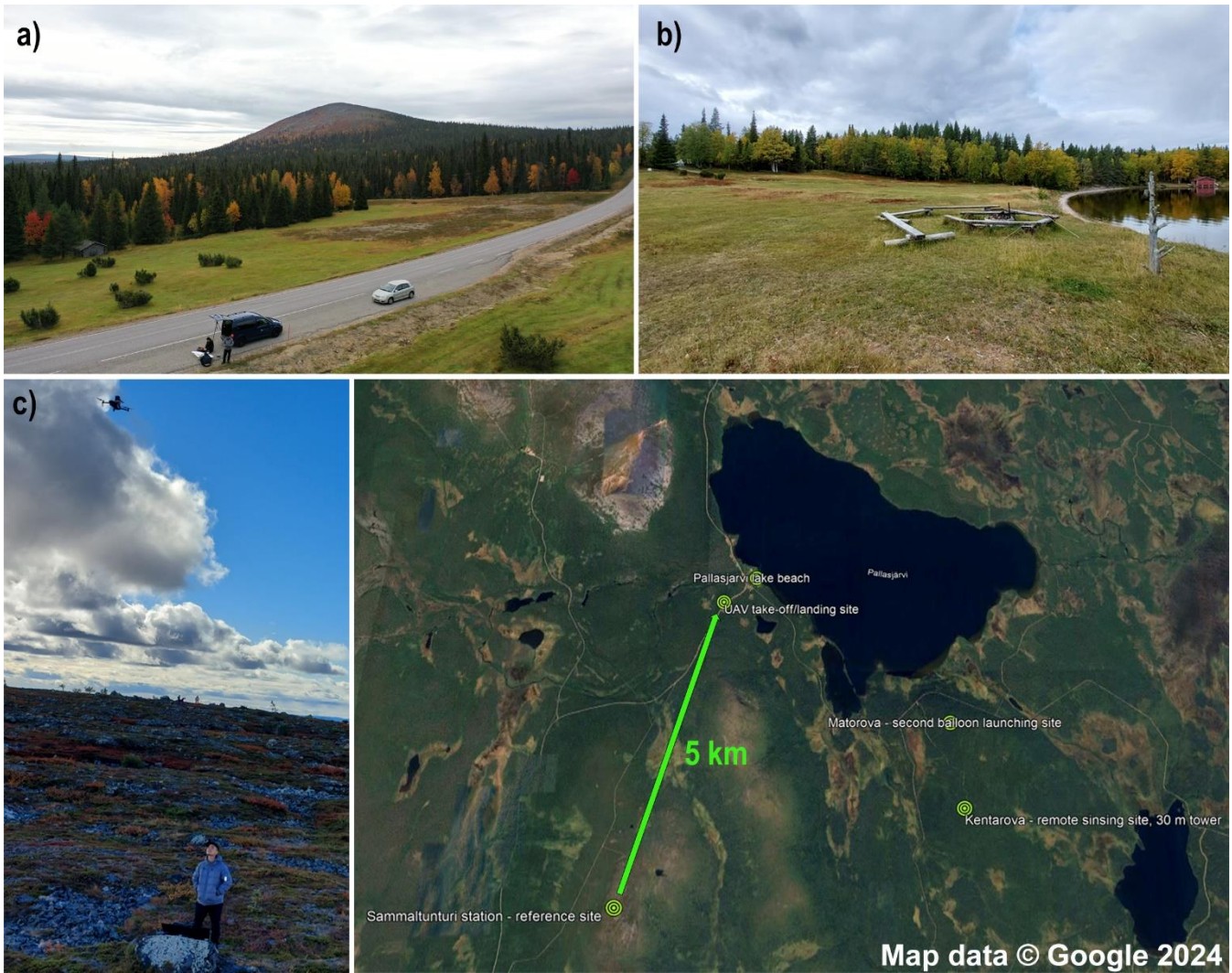

Figure 2. A map showing locations of sampling sites: a) the UAV take-off/landing site next to main road 957 Pallaksentie

115    (68°1'10.30"N 24°8'57.84"E, 304 m ASL, b) the Pallasjarvi lake beach (68°01'23.2"N 24°09'48.8"E, 276 m MSL),

c) calibration flight on the top of hill next to Sammaltunturi station (67°58'24.0"N 24°06'56.3"E, 560 m ASL). Background

map courtesy of © Google Maps.

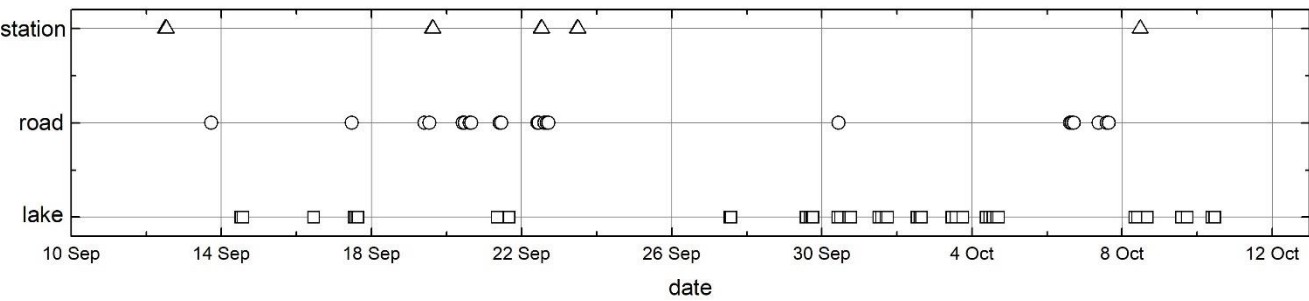

Figure 3. The timeline of DJI Mavic 2 drone flights frequency at three different locations.

Figure 4 provides the spatiotemporal variability in measured meteorological variables and particle number concentration during the IOP of PaCE 2022 campaign. The measured temperature spans from sub-zero -3°C at 500 m AGL to 15 °C at the surface. The RH and dew point exhibit wide variability throughout the measured column, RH ranging from 35 to close to almost 90 %. Please note the bias in RH when measured against the reference below, Fig. 5b. Particle number concentration ranged from very low values of 0.1 to about 300 cm$^{-3}$. Those concentrations might include both aerosol and cloud particles, in full size range from 0.3 to 40 µm (PSL equivalent) of the sensor, since the aerosol flow of OPC-N3 was not dried.

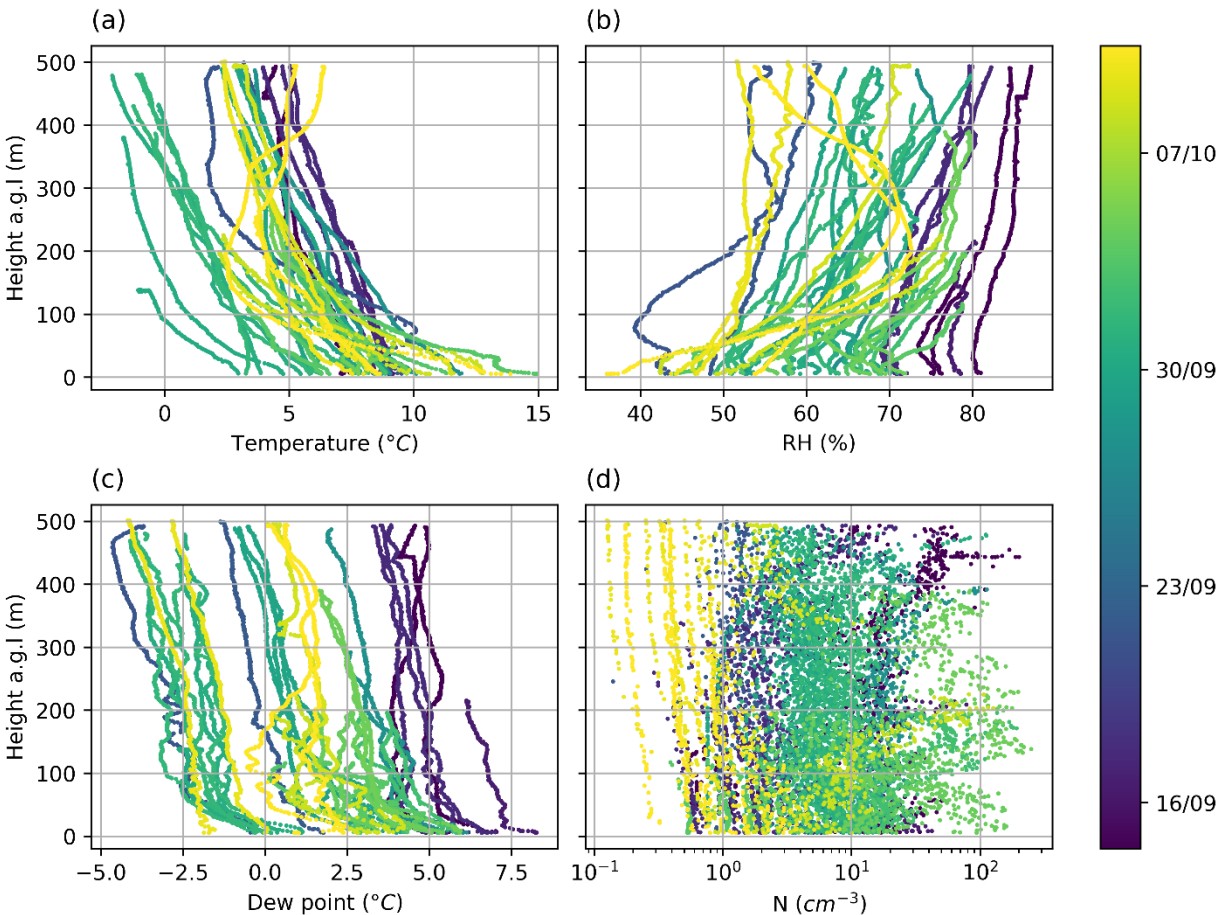

Figure 4. All measurement profiles of meteorological parameters: a) temperature, b) RH, c) dew point and d) particle number concentration (N) collected during the IOP of PaCE 2022. Only ascending parts of profiles are shown. The color scale represents the date of drone operation.

## 4 Data processing and quality control

The data files generated by the drone backpack were formatted in NetCDF and CSV format. Quality control checks applied and missing data points or those with bad values were set to −9999.9. The files were named according to requirements depicted in Brus et al., (2025c), i.e. FMI.DBP.b1.yyyymmdd.hhmm.nc and FMI.DBP.b1.yyyymmdd.hhmm.csv, where DBP stands for drone backpack and yyyymmdd.hhmm corresponds to the date and time of drone take off. These files include the Raspberry Pi real-time clock time stamp (UTC), aircraft location (GNSS latitude and longitude in degrees and altitude in m ASL), meteorological parameters measured by BME280 sensor: temperature (°C), pressure (hPa), and relative humidity (%); meteorological parameters measured by SHT85 sensor: temperature (°C), relative humidity (%), dew point (°C), estimated

wind speed (ms$^{-1}$) and direction (deg). Finally, these files include particle number concentrations (cm$^{-3}$) measured by OPC-N3 sensor in 24 size bins with mid-bin diameters 0.41, 0.56, 0.83, 1.15, 1.5, 2, 2.65, 3.5, 4.6, 5.85, 7.25, 9, 11, 13, 15, 17, 19, 21, 23.5, 26.5, 29.5, 32.5, 35.5 and 38.5 μm, the calculated total particle number concentration (cm$^{-3}$), the calculated d$N$/dlog$D_p$ (cm$^{-3}$) values in each size bin, then measured PM$_1$, PM$_{2.5}$, and PM$_{10}$ mass concentrations (in μg m$^{-3}$). The PM values were calculated by using the Alphasense Ltd. internal algorithm with the default setting for the OPC-N3, the refractive index of 1.5 (real part only) and the density of 1.65 g cm$^{-3}$. Additionally, the files provide temperature (°C), relative humidity (%), sample flow rate (cm$^3$s$^{-1}$) and sampling period (s) of the sample flow, respectively. Since the Raspberry Pi was not connected to the internet, its real-time clock was not synchronized with the GPS clock, causing a slight time lag between some parameters. This issue was resolved by adjusting the time using a lag calculation based on a cross-correlation between the measured pressure by BME280 sensor (1 Hz) and recorded altitude by DJI Mavic 2 pro (10 Hz), we estimate the maximum error in synchronization to be 1 second.

To estimate the uncertainty of the drone backpack measurements, we flown the drone backpack against the reference. We utilized continuous measurements at the Pallas Sammaltunturi station and the 30 m measurement tower of SMEAR 3 station at Kumpula campus in Helsinki. The measurements at Pallas were done during the spring and autumn 2022 to cover as wide temperature and RH ranges as possible. The flights were done next to a 5 m high mast with a 3d anemometer and an automatic weather station (Milos 500, Vaisala Inc.). The measurements in Kumpula, Helsinki were conducted in spring 2022 against the 30 m high meteorological tower, which is equipped with meteorological instrumentation at several heights (4, 8, 16 and 30 m). However, intercomparison was done only at the 30 m level to ultrasonic anemometer (Metek USA-1), platinum resistance thermometer (Pt-100) and thin film polymer sensor (Vaisala DPA500). Further, the aerosol instrumentation for particulate measurements was placed on the FMI building roof which is at the same height (30 m AGL) and about 50 m air distance from the tower. The particulate matter intercomparison measurements were done against two optical particle counters (OPC): the optical particle spectrometer (OPS model 3330, TSI Inc.) with a size range of 0.3-10 μm and the mini cloud droplet analyzer (mCDA, Palas GmbH) with a size range of 0.35-17.6 μm. The drone hovered between the meteorological tower and FMI building in Kumpula campus. The horizontal inlet of OPC-N3 placed on top of the drone backpack was always facing the wind direction to minimize the sampling losses due to non-isokinetic sampling (Julaha et al., 2025). The inlets of both reference OPCs were oriented vertically. The measurements followed a consistent pattern: the drone took off from the ground and ascended to the height of the reference measurements. The drone was then orientated so its front was heading into the wind. The drone in GPS attitude flight mode was hovering next to reference (about 5 m horizontal distance) for at least 10 minutes at low and about 20 minutes at higher ambient temperatures, respectively. Data from both the drone backpack and reference instruments were averaged over the whole stably hovering period.

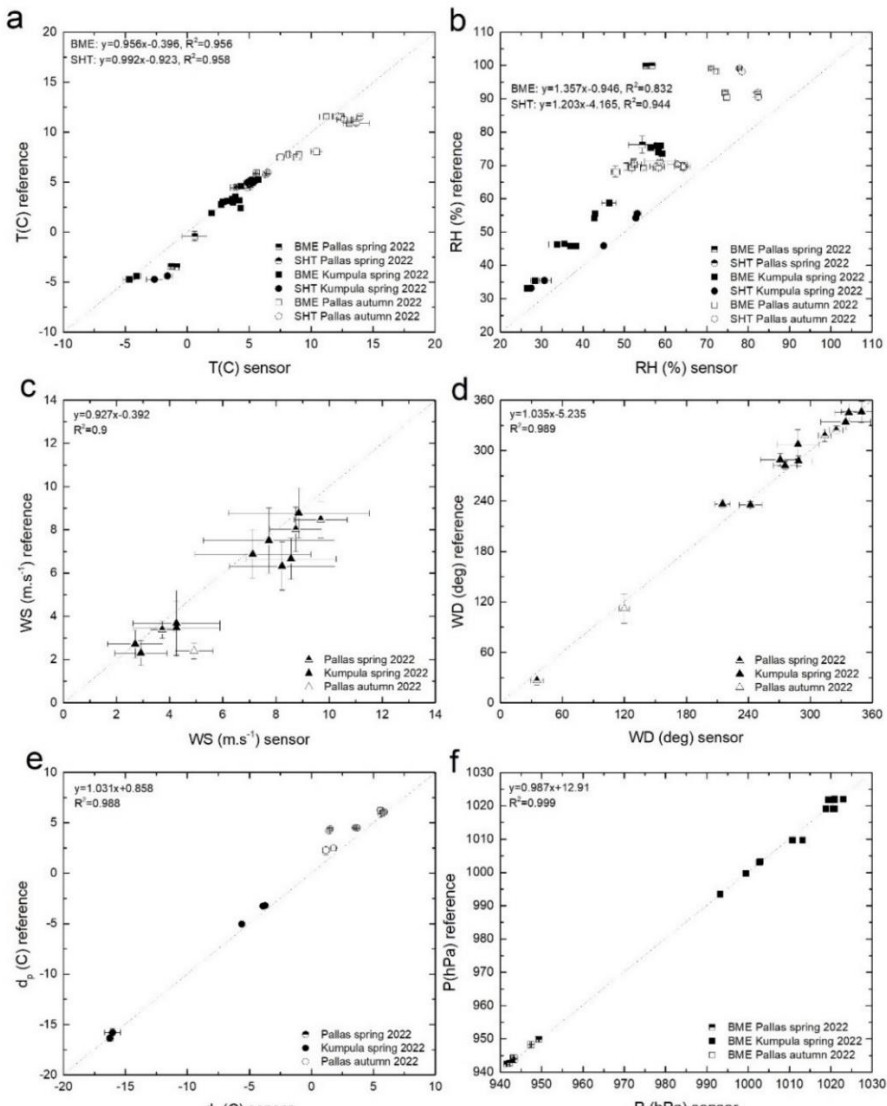

Figure 5. Drone backpack measurements against the reference at given altitude, panel a) temperature, b) relative humidity, c) wind speed, d) wind direction, e) dew point, f) pressure. The dotted line represents the perfect agreement between drone backpack and reference instrumentation; it is included only to lead the readers' eyes. The linear fit with R-squared value is depicted in each panel separately.

Figure 5 presents the intercomparison of meteorological variables measurements for a) temperature, b) RH, c) wind speed, d) wind direction, e) dew point, f) pressure. The linear fit and R-squared values are provided in each panel for each sensor separately. The drone backpack sensors' temperature measurements tend to slightly overestimate the reference measurements.

The relative humidity measurements are having the highest uncertainty (similarly as in Barbieri et al., 2019), some measurements were done at very high ambient RH and the sensors became saturated with water vapor and condensation occurred on sensor's surface. The third-party wind estimates from drone attitude seem to be catching wind direction more accurately than the wind speed. The wind speed estimate has a positive bias about 1 m.s$^{-1}$, but up to maximum of 2 m.s$^{-1}$. It must be noted that it was not possible to get the wind speed and direction estimates for all flights due to failure in saving the drone attitude data to drone flight controller. The problem was encountered in 10 out of 14 in case of Sammaltunturi station and 7 out of 16 in the case of Kumpula reference flights. The dew point and pressure measurements follow the reference satisfactorily. However, it should be noted that the published dataset is at level b1 (i.e. data with quality control checks applied and missing data points or those with bad values were set to −9999.9) with no calibration factors applied, as specified in Brus et al., (2025c).

Figure 6 panel a) depicts the total particle concentration intercomparison of 8 flights conducted in the first week of March 2022 and panel b) the particle number size distribution. The particle number concentration is generally very low for the measured size range, up to 10 cm$^{-3}$. The variation in particle concentration is naturally higher for OPC-N3 mounted on top of the drone backpack. We believe this is due to external forces, like gust wind or sudden changes in wind direction, impacting the drone attitude and thus the particulate measurements. The particle number size distributions (only shown for 3 consecutive flights on March 2$^{nd}$, 2022) of all OPCs follow the same shape and the uncertainty in counting among the OPCs is about factor of 2 which is not unusual for drone aerosol measurements, see e.g. Brus et al., (2021). Similarly, as for meteorological variables the particulate measurements dataset is at level b1 with no calibration factors applied.

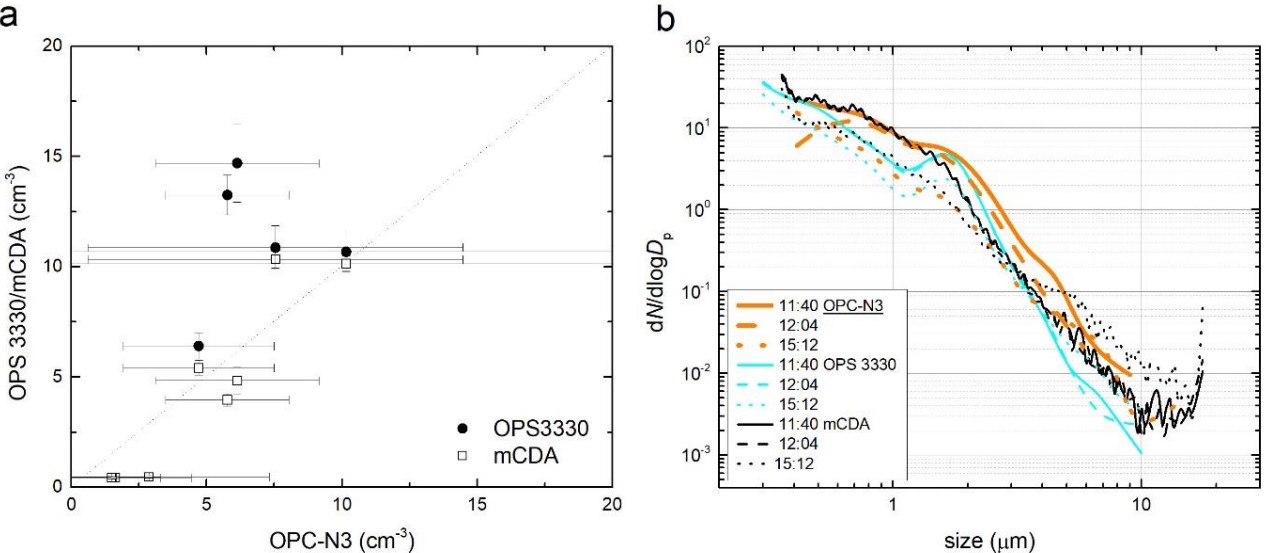

Figure 6. Drone backpack OPC-N3 particulate matter measurements against reference instruments OPS (model 3330, TSI Inc.) and the mCDA (Palas GmbH). Panel a) total aerosol concentration, b) particle number size distribution, both measured within the range of each instrument.

**Dataset remarks**

Since hysteresis in T and RH profiles collected by FMI drone backpack was noticeable, we recommend using only data from the ascending portion of the flown profiles. The descent data were found to be biased due to washout of descending drone's propellers and exhibiting rather flat profile.

## 5 Data availability

Datasets collected by FMI drone backpack were published at the Zenodo Open Science data archive (http://zenodo.org, last access: 5 February 2025) under a dedicated community Pallas Cloud Experiment – PaCE2022 (https://zenodo.org/communities/pace2022/, last access: 5 February 2025) as NetCDF and CSV archives: https://doi.org/10.5281/zenodo.14780929 (Brus et al., 2025a) and https://doi.org/10.5281/zenodo.14778421 (Brus et al., 2025b), respectively.

## 6 Code availability

The Python scripts developed to log, process and display data are not publicly available, however could be obtained from authors on request for free.

## 7 Summary

This manuscript provides measurements and datasets collected by the FMI during Pallas Cloud Experiment (PaCE 2022) campaign. The campaign took place in norther Finland during the autumn of 2022. In Section 2, we provided an overview of the platform deployed during this campaign and offered the payload description – the custom-built drone backpack for air quality and atmospheric state variables carried on top of the consumer-grade drone (DJI Mavic 2 pro). In Section 3 we described the flight strategies, while Section 4 provided an overview of the datasets obtained, including a description of the measurement against the reference for data validation. Section 5 provided information on the dataset's availability, all files are available in both netCDF and CSV format.

During the PaCE 2022 campaign different airborne platforms, including UAVs, and tethered balloon systems (TBS), carrying various payloads for in-situ aerosol and cloud physical properties and atmospheric state variables measurements were deployed concurrently. Also, continuous surface in-situ measurements of aerosol and cloud properties are available from the

Sammaltunturi hilltop station, those could serve as a reference or as complementary to further analysis. We encourage prospective users to integrate the drone backpack measurements with the comprehensive dataset of aerosol physical and optical properties from the hilltop station, as summarized by Backman et al. (2025). Specifically, the online ice-nucleating particle (INP) measurements presented by Böhmländer et al. (2025a) and fluorescent aerosol measurements by Gratzl et al. (2025) offer a valuable complement. Further analysis and inter-comparison of various sensor data can be conducted against other airborne measurements. These include fixed-wing UAV aerosol and cloud in-situ measurements by Girdwood et al. (2025), UAV INP profiling by Böhmländer et al. (2025b), and tethered balloon system (TBS) measurements covering turbulence and cloud microphysics by Schlenczek et al. (2025). Additionally, high-resolution TBS profiling of the boundary layer by Chavez-Medina et al. (2025) and aerosol and cloud measurements by Le et al. (2025) provide further avenues for comparative studies. Moreover, aerosol properties below the cloud base can be analyzed using lidar backscatter, aerosol depolarization ratio, and turbulence parameters derived from the remote sensing dataset presented by Tukiainen et al. (2025). All the datasets from the "Data generated during the Pallas Cloud Experiment 2022 campaign" special issue of ESSD provide a comprehensive foundation for researchers investigating aerosol-cloud interactions and their dynamics.

**Author contributions**

DB planned and coordinated the FMI flights during PaCE 2022 campaign, DB, VL and KD conducted drone backpack measurements, VL and DB processed, analyzed, and quality-controlled FMI dataset. JK and DB designed the drone backpack. All authors contributed to the writing of the manuscript and quality controlled the FMI dataset.

**Competing interests**

The authors declare that they have no conflict of interest.

**Special issue statement**

This article is part of the special issue:" Data generated during the Pallas Cloud Experiment 2022 campaign".

**Acknowledgements**

The authors would like to acknowledge Metsa halitus personnel, namely Mirka Hatanpää, for countless support during Pallas Cloud experiment 2022.

**Financial support**

255 This work was supported by ACTRIS-Finland funding through the Ministry of Transport and Communications, the Atmosphere and Climate Competence Center Flagship funding by the Research Council of Finland (Grants 337552). This project has also received funding from the European Union, H2020 research and innovation program (ACTRIS-IMP, the European Research Infrastructure for the observation of Aerosol, Clouds, and Trace gases, Grant 871115).

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
