# Peer review of "Data collected by a drone backpack for air quality and atmospheric state measurements during Pallas Cloud Experiment 2022 (PaCE2022)"

_Earth System Science Data, 2025_

## Author Comment (AC1)

We would like to thank both reviewers for their time and suggestions to improve our manuscript, we really appreciate their effort. Below can be found responses to reviewers' comments as RC - reviewer comment and AA – authors answer.

**Reviewer 1**

This study presents an extensive dataset collected using a drone-based backpack system for air quality and atmospheric state measurements during the Pallas Cloud Experiment 2022 (PaCE2022). The authors detail the instrumentation, calibration/validation, and data collection methods used during the campaign, emphasizing the advantages of drone-based measurements for atmospheric studies, especially in subarctic areas. The dataset includes information on aerosol concentrations, and meteorological parameters, providing insights into the atmospheric conditions of the studied area.

**Strong points:**

The dataset has been rigorously validated through comparisons with reference measurements, which enhances the credibility and usability of the collected data.

The dataset offers valuable information for the atmospheric science community, particularly regarding the use of UAV-based measurement techniques in complex and/or under-studied atmospheric conditions.

**Suggested improvements:**

RC: While the introduction provides a solid background on the importance of UAV-based measurements, it lacks a clear structure that outlines the research objectives and the organization of the paper. Providing a more structured introduction would enhance readability and help guide the reader through the study.

AA: the following paragraph was added to introduction: "Our dataset offers a unique opportunity for the broader scientific community to better understand the vertical structure of near-surface aerosol particles in a subarctic environment, revealing their crucial role in influencing low-level stratiform cloud microphysical and radiative properties. In Section 2, we describe the drone measurement platform, the assembly of the backpack module with all sensors and their operational characteristics. Section 3 details the measurement sites, flight strategy and presents the completed vertical profile measurements. Section 4 explains the dataset structure, quality control and assurance of data. Section 5 provides direct links to Zenodo dataset repository with netCDF and CSV files."

RC: Although the dataset is well-documented, the discussion on its potential applications and future uses is relatively limited. Expanding the conclusion to explicitly address how

this dataset could be utilized by the scientific community and integrated into broader atmospheric research (e.g., CCN, INP) would strengthen the impact of the study.

AA: The following paragraph was added to "Summary section": "We encourage prospective users to integrate the drone backpack measurements with the comprehensive dataset of aerosol physical and optical properties from the hilltop station, as summarized by Backman et al. (2025). Specifically, the online ice-nucleating particle (INP) measurements presented by Böhmländer et al. (2025a) and fluorescent aerosol measurements by Gratzl et al. (2025) offer a valuable complement. Further analysis and inter-comparison of various sensor data can be conducted against other airborne measurements. These include fixed-wing UAV aerosol and cloud in-situ measurements by Girdwood et al. (2025), UAV INP profiling by Böhmländer et al. (2025b), and tethered balloon system (TBS) measurements covering turbulence and cloud microphysics by Schlenczek et al. (2025). Additionally, high-resolution TBS profiling of the boundary layer by Chavez-Medina et al. (2025) and aerosol and cloud measurements by Le et al. (2025) provide further avenues for comparative studies. Moreover, aerosol properties below the cloud base can be analyzed using lidar backscatter, aerosol depolarization ratio, and turbulence parameters derived from the remote sensing dataset presented by Tukiainen et al. (2025). All the datasets from the "Data generated during the Pallas Cloud Experiment 2022 campaign" special issue of ESSD provide a comprehensive foundation for researchers investigating aerosol-cloud interactions and their dynamics.

**Minor comments line by line:**

RC: L10: "against the reference" - Please explain the meaning.

AA: Authors meant :" ...and 12 inter-comparison flights against the reference instrumentation at Sammaltunturi station." The text will be changed accordingly.

RC: L11: "meteorological parameters" - Which ones?

AA: the text was updated as follows: "... and meteorological parameters (temperature, relative humidity, pressure, wind speed and direction) up to 500 m above the ground level."

RC: L12-14 - The provided links include the coma at the end, thus are not working when direct click on them.

AA: links were corrected.

RC: L24 - "our previous research" - Please specify and cite.

AA: The references are provided in lines 26 and 27.

RC: L42 - Also check https://doi.org/10.5194/amt-2024-162. Can be also interesting to consider as this study used the same OPC in a different drone system to study volcanic aerosols.

AA: Authors would have several objections on the mentioned manuscript especially the design of aerosol sampling e.g. efficiency and non-isokinetic sampling of vertically orientated inlet. However, the authors will add this study as yet another example of use of low-cost OPC on UAV.

RC: L42 - "FMI" - Please specify it properly the first time for people that does not know the Finish Meteorological Institute.

AA: Corrected.

RC: L59 - "minimize the propeller airflow" - Based on what ? You could add some references that indeed show propeller airflow is minimal in this drone area (e.g., https://doi.org/10.2514/6.2018-1266, https://doi.org/10.2514/1.C032828, https://doi.org/10.3390/drones6110329).

AA: Modelling studies are very important for initial estimates, but they provide only theoretical answers to in advance well defined problems. There are many CFD simulations describing the air flow around various UAV configurations, several deal with influence of propeller downwash on aerosol deposition (e.g. agricultural sprayers), but close to none include aerosol dynamics in wide range of sizes above the propeller plane. Authors will include reference of Ghirardelli et al., (2023) that describes in detail the flow structure around multicopter drone.

Our lab has over 15 years of hands-on experience deploying aerosol instrumentation on a variety of airborne platforms, including both manned and unmanned aircraft. Every new aerosol sampling design we develop undergoes thorough field testing against reference instruments, a process that inherently involves numerous failures and iterative improvements. Unfortunately, due to limited resources, we aren't able to conduct CFD simulations for every new sampling design.

RC: L110 - Might be useful to specify here the size range of the measured particles.

AA: The sentence was change as follows: "Those concentrations might include both aerosol and cloud particles, in full size range from 0.3 to 40 µm (PSL equivalent) of the sensor, since the aerosol flow of OPC-N3 was not dried."

RC: L128 - I suppose that PM are calculated from the raw particle counts of the OPC, but based on which particle density?

AA: The PM values were calculated by using the Alphasense internal algorithm with the default setting for the OPC-N3, the refractive index of 1.5 (real part) and the density of 1.65 g cm−3. That value corresponds to the typical range of densities for various types of airborne particulate matter and is considered as a reasonable compromise.

RC: Figure 5 - Not really clear why you have such errors/uncertainties on the OPC measurements.

AA: The error bars on x-axis (drone backpack) are indeed greater in magnitude. We believe it is due to external forces impacting the drone attitude and thus the particulate measurements, like gust wind or sudden changes in wind direction. In figure 1, the correlation of error bars magnitude and wind speed is evident.

[Figure]

Figure 1. Drone backpack OPC-N3 particulate matter measurements against reference instruments OPS (model 3330, TSI Inc.) and the mCDA (Palas GmbH) on the left-side panel accompanied with wind speed measurements on the right-side panel.

The sentence will be restated as follows: "The variation in particle concentration is naturally higher for OPC-N3 mounted on top of the drone backpack. We believe this is due to external forces, like gust wind or sudden changes in wind direction, impacting the drone attitude and thus the particulate measurements."

**References:**

Backman, J., Luoma, K., Servomaa, H., Vakkari, V., and Brus, D.: In-situ aerosol measurements at the Arctic Sammaltunturi measurement station during the Pallas Cloud Experiment 2022, Earth Syst. Sci. Data Discuss. [preprint], essd-2025-284, in review, 2025.

Böhmländer, A., Lacher, L., Fösig, R., Büttner, N., Nadolny, J., Brus, D., Doulgeris, K.-M., and Möhler, O.: Measurement of the ice-nucleating particle concentration with the Portable Ice Nucleation Experiment during the Pallas Cloud Experiment 2022, Earth Syst. Sci. Data Discuss. [preprint], https://doi.org/10.5194/essd-2025-89, in review, 2025a.

Böhmländer, A. J., Lacher, L., Höhler, K., Brus, D., Doulgeris, K.-M., Girdwood, J., Leisner, T., and Möhler, O.: Measurement of the ice-nucleating particle concentration using a mobile filter-based sampler on-board of a fixed-wing uncrewed aerial vehicle during the Pallas Cloud Experiment 2022, Earth Syst. Sci. Data Discuss. [preprint], https://doi.org/10.5194/essd-2025-87, in review, 2025b.

Ghirardelli, M., Kral, S.T., Müller, N.C., Hann, R., Cheynet, E., Reuder, J. Flow Structure around a Multicopter Drone: A Computational Fluid Dynamics Analysis for Sensor Placement Considerations. Drones, *7*, 467, https://doi.org/10.3390/drones7070467, 2023.

Girdwood, J., Brus, D., Doulgeris, K., Böhmländer, A.: Data From the Universal Cloud and Aerosol Sounding System Abord an Uncrewed Aircraft During the Pallas Cloud Experiment 2022, Earth Syst. Sci. Data Discuss. [preprint], essd-2025-257, 2025.

Gratzl, J., Brus, D., Doulgeris, K., Böhmländer, A., Möhler, O., and Grothe, H.: Fluorescent aerosol particles in the Finnish sub-Arctic during the Pallas Cloud Experiment 2022 campaign, Earth Syst. Sci. Data Discuss. [preprint], https://doi.org/10.5194/essd-2024-543, in review, 2025.

Chávez-Medina, V., Khodamoradi, H., Schlenczek, O., Nordsiek, F., Brunner, C. E., Bodenschatz, E., and Bagheri, G.: Max Planck WinDarts: High-Resolution Atmospheric Boundary Layer Measurements with the Max Planck CloudKite platform and Ground Weather Station – A Data Overview, Earth Syst. Sci. Data Discuss. [preprint], https://doi.org/10.5194/essd-2025-111, in review, 2025.

Le, V., Doulgeris, K. M., Komppula, M., Backman, J., Bagheri, G., Bodenschatz, E., and Brus, D.: Dataset of airborne measurements of aerosol, cloud droplets and meteorology by tethered balloon during PaCE 2022, Earth Syst. Sci. Data Discuss. [preprint], https://doi.org/10.5194/essd-2025-148, in review, 2025.

Schlenczek, O., Nordsiek, F., Brunner, C. E., Chávez-Medina, V., Thiede, B., Bodenschatz, E., and Bagheri, G.: Airborne measurements of turbulence and cloud microphysics during PaCE 2022 using the Advanced Max Planck CloudKite Instrument (MPCK[+]), Earth Syst. Sci. Data Discuss. [preprint], https://doi.org/10.5194/essd-2025-112, in review, 2025.

Tukiainen, S., Siipola, T., Leskinen, N., and O'Connor, E.: Remote sensing measurements during PaCE 2022 campaign, Earth Syst. Sci. Data Discuss. [preprint], https://doi.org/10.5194/essd-2024-605, in review, 2025.

---

## Author Comment (AC2)

We would like to thank both reviewers for their time and suggestions to improve our manuscript, we really appreciate their effort. Below can be found responses to reviewers' comments as RC - reviewer comment and AA – authors answer.

**Reviewer 2**

This data description paper is a very important addition to the already published data from the Pallas Cloud Experiment 2022. It contains information about the measurement system, data acquisition, measurement strategy and some intercomparison with other instruments. It is good to see researchers presenting their methods and instruments alongside published datasets!

Most information is sufficient to understand and further use the data, including its peculiarities, but especially for the csv-Files with its high accessibility, details are needed for people not being part of the experiment to make most out of the data.

Although I don't feel qualified to make comments on the language, the text might benefit a lot from using a language tool or a native speaker, in addition to some basic decisions (data as singular or plural? Direct or indirect speech?) to be followed throughout the text.

In the following, you'll find some specific comments with line numbers:

RC: L17: "..all [atmospheric?] models"; add a reference to the statement?

AA: The word "atmospheric" and the reference to Morrison et al, (2020) were added.

RC: L47: please omit the word "please"

AA: word "please" was omitted.

RC: L55: Why is it suitable (include a reference for the statement)? Or is it intended to be used for .. ?
AA: The wording was changed according to reviewer suggestion to " It is intended to be used for air quality monitoring…". And Figures 4 and 5 of this manuscript, in our opinion, also suggest that it is suitable for such tasks.

RC: L62: rather mention the relative airflow in the drone coordinate system than wind speed? E.g. a racetrack with the wind would decrease the relative airflow (and therefore affect the sensor ventilation).

AA: In our operation, the drone is moving strictly in vertical column always heading into wind, thus there is no relative movement of the drone in horizontal axis, except for cases with very high winds (>15 m/s), when the drone slides from desired coordinates. The sentence was changed accordingly: "…the aspiration rather depends on drone vertical move and horizontal

airflow for the air exchange around the sensors."

RC: L63-70: from my perspective, there are too many details not relevant for working with the data (for example it is not important to name the interfaces used for the sensors, e.g. I2C/SPI/Serial/..), please consider shortening this paragraph.

AA: The technical details were omitted according to reviewers' suggestions.

RC: L72-75: Is the source code available on git? Maybe describe some more your wind estimation algorithm to allow the user to estimate its strength and weakness?

AA: Unfortunately, the code is not available. As stated in manuscript on L74 "A proprietary wind algorithm version 2.2 and Mavic 2 pro aerodynamic profile were used for wind estimates within this manuscript."

RC: Table 1: This is a good starting point to show the manufacturers estimates on resolution/accuracy/uncertainty and response time; maybe fill up the values not provided in datasheets with your estimations? E.g. Res/Acc for OPC, Vertical positioning for GNSS. I somehow missed a concluding table with your guess on resolution/accuracy/uncertainty and response time of your whole measurement system, which likely will achieve slightly less accurate measurements on a drone than in the lab. In addition, isn't the response time of RH temperature dependent? If so, please make a comment (at least reference temperature for the response time).

AA: Table 1 was updated with OPC-N3 resolution in PM fractions; however the accuracy estimates are highly influenced by ambient conditions e.g. high humidity and extreme temperatures. We aimed to provide an overview on accuracy and uncertainty of all sensors with our measurements against the reference instrumentation at the Sammaltunturi station and Kumpula campus, Figures 4 and 5, with included linear fits and coefficients of determination.

RC: L84..86: Can you provide some links to the mentioned networks (ACTRIS/ICOS/..) and mention implications/benefits for the station and its measurements?

AA: Links to research programs at Pallas supersite ( https://en.ilmatieteenlaitos.fi/pallas-atmosphere-ecosystem-supersite) were added to text. ICOS (https://www.icos-cp.eu/), ACTRIS (https://www.actris.eu/) and EMEP (https://emep.int/). Also reference to overview manuscript research programs at Pallas by Lohila et al. (2015) was added.

RC: L87ff: A map (although referred to a map in another publication later in the text) and especially a picture of the sites and conditions during the experiment could help a lot to

understand data and the general environment (snow/grass/flora), including the low level clouds during fall.

AA: Figure 2 was added to manuscript as suggested.

[Figure]

Figure 2. A map showing locations of sampling sites: a) the UAV take-off/landing site next to main road 957 Pallaksentie (68°1'10.30"N 24°8'57.84"E, 304 m ASL. b) the Pallasjarvi lake beach (68°01'23.2"N 24°09'48.8"E, 276 m MSL). c) Calibration flight on the top of hill next to Sammaltunturi station (67°58'24.0"N 24°06'56.3"E, 560 m ASL). Background map courtesy of © Google Maps.

Also, there is a dedicated YouTube playlist from PaCE2022, that provides insight into environment and conditions during the campaign. All that information is included in our PaCE2022 Overview manuscript, that we were advised by editors to publish as last in this special issue.

https://www.youtube.com/playlist?list=PLK2ec25bvC9PddfM9ezMyhjbaU7B1MMfc

RC: L100: consider omitting the text about programming the mission

AA: the sentence was restated to: "The flight missions were conducted by using DJI GO 4 software, and both ascent and descend rates were set the same to 1 m s$^{-1}$."

RC: L110: add a reference to the picture/section where one can see the RH bias?

AA: The reference to Figure 5 b was added.

RC:L122: please explain the altitude further - is it mean sea level in addition to a specific geoid (e.g. EGM96)? Consider using GNSS instead of GPS.

AA: The positioning data are taken from DJI flight records. DJI drones rely on a barometric altimeter, this sensor measures air pressure to determine changes in altitude relative to the take-off point. The barometric altimeter is crucial for flight control, it's not a geodetic measurement and is susceptible to atmospheric pressure changes. The reported altitude is not the same as a geoid-derived orthometric height. The actual atmospheric pressure varies significantly with weather, temperature, and local conditions. So, while it's "above sea level" in the sense of a standard atmospheric model, it's not a precise geodetic Mean Sea Level (MSL) as defined by a geoid.

The GPS was changed to GNSS as suggested.

RC: L132: Good to read how the data was synchronized. You might add a short note of the error you expect in the time synchronization.

AA: Similarly as above, the sensors' data (1Hz) are synced with DJI flight records (10 Hz), we expect maximum error in sync of 1 sec. The sentence was changed as follows: "This issue was resolved by adjusting the time using a lag calculation based on a cross-correlation between the measured pressure by BME280 and recorded altitude by DJI Mavic 2 pro, we estimate the maximum error in synchronization to be 1 second."

RC: L146: consider adding a reference about sampling losses.

AA: The reference about sampling losses calculations for OPC-N3 on drone could be found in Supplementary Materials of Julaha et al. 2025. The reference was added.

RC: L168: please add a reference/table for the dataset levels (b1 here) so the reader is able to understand it.

AA: The sentence was changed as follows:"... the published dataset is at level b1 (i.e. data with quality control checks applied and missing data points or those with bad values were set to −9999.9) with no calibration factors applied, as specified in Brus et al., (2025c).

L195: is all information about the processing steps (some nonlinear corrections / wind estimation algorithm and parameters / .. ) present in the publication to allow dataset users to understand the (pre-)processed data?

Regarding dataset users:

RC: For the nc-Files, metadata description within the file is clear (although no instrument is mentioned in the variable attributes for e.g. particle concentration), but for the CSV file, more information within this data description paper would be helpful (e.g. bin numbering and bin size/edges in a table, not just within the text[L127]). A concluding table with estimated overall uncertainty, response time and e.g. repeatability for each variable in your datasets might strongly increase the ability of dataset users to work and publish with the provided data.

This publication is a very important addition to the provided datasets, once all (or at least most of) the meta-information is well presented!

AA: We agree that concluding table with estimates overall uncertainty, repeatability for each variable would be of great benefit. However, only close by reference at the sampling places were other platforms, tethered balloons and UAVs, their data sets will be available in this special issue. We plan to do rigorous analysis combining all platforms and available data sets in this special issue.

The metadata csv file was uploaded to Zenodo database of this data paper repository. The metadata file contains: Bin low boundary (particle diameter [um]), Bin mean (particle diameter [um]), Volume of a particle in bin (um^3) and Weighting for each bin.

References:

Morrison, H., van Lier-Walqui, M., Fridlind, A. M., Grabowski, W. W., Harrington, J. Y., Hoose, C., Korolev, A., Kumjian, M. R., Milbrandt, J.A., Pawlowska, H., Posselt, D. J., Prat, O. P., Reimel, K. J., Shima, S.-I., van Diedenhoven, B., Xueet L.: Confronting the challenge of modeling cloud and precipitation microphysics. Journal of Advances in Modeling Earth Systems, 12, e2019MS001689. https://doi.org/10.1029/2019MS001689, 2020.

Lohila A., Penttilä T., Jortikka S., Aalto T., Anttila P., Asmi E., Aurela M., Hatakka J., Hellén H., Henttonen H., Hänninen P., Kilkki J., Kyllönen K., Laurila T., Lepistö A., Lihavainen H., Makkonen U., Paatero J., Rask M., Sutinen R., Tuovinen J.-P., Vuorenmaa J. & Viisanen Y. 2015. Preface to the special issue on integrated research of atmosphere, ecosystems and environment at Pallas. Boreal Env. Res. 20, 431–454.

Julaha, K., Ždímal, V., Mbengue, S., Brus, D., and Zíková, N.: Drone-based vertical profiling of particulate matter size distribution and carbonaceous aerosols: urban vs. rural environment, EGUsphere [preprint], https://doi.org/10.5194/egusphere-2025-1420, 2025.